## [Transparent Peer Review file · Nature Communications]

Adversarial Attacks on Large Language Models in Medicine

Corresponding Author: Dr Zhiyong Lu

Version 0:

Reviewer comments:

Reviewer #1

(Remarks to the Author)

Having reviewed both the original and revised manuscript, I find that the authors have made substantial improvements addressing most of the previously raised concerns. The paper now presents a more rigorous and comprehensive analysis of LLM vulnerabilities in medical applications. The revised version demonstrates significant improvements in several key areas. The methodology and terminology have been clarified considerably, making the research more accessible and reproducible. The quantitative evaluation has been strengthened with the addition of confidence intervals and comprehensive statistical analysis. The expansion of model coverage to include various LLM variants enhances the generalizability of the findings. The authors have successfully demonstrated the specific vulnerabilities in medical tasks, providing valuable insights for the healthcare community. Furthermore, their analysis of the relationship between poisoned data proportion and attack effectiveness offers novel contributions to the field. However, there are two areas where the manuscript could be further enhanced.

1) Regarding defense mechanisms, while the authors have introduced weight scaling as a potential defense, there are other promising approaches that could be explored. In particular, paraphrase-based detection methods could be particularly effective in the medical domain, where standardized terminology is common. Such methods could leverage the structured nature of medical language to detect potential attacks by comparing model outputs across different phrasings of the same input.

2) The manuscript would benefit from a more thorough discussion of how model updates might affect the discovered vulnerabilities. While the current version acknowledges this limitation, a deeper exploration of how regular model updates might impact attack effectiveness would be valuable. This could include theoretical considerations of how architectural changes might enhance model robustness, particularly in the context of medical applications.

Despite these suggestions for improvement, I believe the manuscript makes a significant contribution to understanding LLM vulnerabilities in medical applications. The findings have important implications for the deployment of LLMs in healthcare settings and will be of significant interest to the broader AI safety and medical informatics communities.

(Remarks on code availability)

Reviewer #2

(Remarks to the Author)

The authors have addressed all of my comments from the previous submission. There would be some possible improvements, but due to the fast pace of this field, I think it would be interesting for the community to share this article with the readership of Nature Communications now.

(Remarks on code availability)

Reviewer #3

(Remarks to the Author)

Thank you to the authors for responding to the feedback during the review process. Some major concerns about the manuscript still persist below. Moreover, based on some of the feedback from R3, the rebuttal was a great opportunity to include more advanced and cutting-edge models (gpt-4o/o1, Claude 3.5, Gemini 2, Llama 3, etc). Unfortunately for the authors, the field is moving very rapidly and the models that were used (especially PMC-Llama, vicuna, etc) are quite dated and may not reflect to the true capabilities of contemporary LLMs.

R2C2: It is still unclear that adding conflicting instructions to a prompt counts as an adversarial attack. Adversarial attacks are commonly regarded to be imperceivable variations added to the input of a model. The authors mention that for custom GPT applications, prompts can be hidden from the end users. However, custom GPTs were not used in the study and simple API-based prompting models were used, where the prompt is visible to the end users. Consequently, it is not apparent that the modes described here are adversarial attacks to a model.

R2C3: Thank you for trying to incorporate how some of the LoRA work can be used to mitigate the proposed attacks. In looking at Figure 5, it looks like it is challenging to distinguish the three histogram of L_{∞} weights. Moreover, in Figure 6, it does not seem like there are any clear conclusions that can be drawn. Yes, the ultrasound ASR rate drops to 25%, but that is not replicated for the other tasks and there is no principled analysis/takeaway that can unfortunately be drawn from these experiments.

R2C5: The impact of summarization was not evaluated (there is substantial prior work that describes the variable quality and approaches of summarizing clinical notes). Moreover, reducing to average token length to <700 when the shortest context model can handle sequence lengths $>4k$, seems to be limiting the utility of the model. Evaluating the maximal quality of an LLM would be a fair approach, instead of reducing to the prompt to the least common denominator.

R2C8: What is the selection bias that is created in using notes >1000 words? How many patients have to be filtered out in the first place? These patient sampling factor are important and may have a large role in understanding the results.

(Remarks on code availability)

I reviewed the code but did not execute it. The readme seems sufficient. A majority is a monolithic Jupyter notebook and not modular components.

Version 1:

Reviewer comments:

Reviewer #1

(Remarks to the Author)

The authors improved the manuscript according to the reviewers' comments and suggestions. The manuscript meets the criteria for publication. I believe the manuscript makes a significant contribution to understanding LLM vulnerabilities in medical applications.

(Remarks on code availability)

Reviewer #3

(Remarks to the Author)

NCOMMS-24-86106A

Thank you to the authors for being responsive to several of the comments that have been brought up during the review process. Most of my comments have been addressed, but a few important comments still remain as described below.

Comment 1:

Thank you for adding some of the latest Generation Frontier models. I believe that this work will make it more relevant for the broader readership of the journal.

Comment 2:

I agree with the response that is provided about how such prompt injection attacks may affect custom GPTs. The response that was included in the rebuttal should also be included in the discussion so that the readers have broader awareness of this topic. Furthermore, the abstract should also explicitly describe what are the adversarial perturbations that are introduced, specifically "prompt injections with malicious instructions" and "poison sample fine-tuning". This will make the abstract better reflect the work. Moreover, if the editorial team agrees, a more representative (albeit) title could be "Adversarial Attacks via Malicious Prompt Injections and Fine-Tuning Data on Large Language Models in Medicine" (or similar). Given the multiplicity of definitions for adversarial robustness, many of which are not covered in this particular manuscript, being specific about what is indeed covered would be helpful.

Consequently, the final statement of the abstract could also be rephrased to be less alarmist because of the somewhat artificial setting of the adversarial attacks used in this study (since zero-shot applications of such methods without malicious

prompting or fine-tuning do not encounter safety risk).

Comment 3:

Thank you for adding the KDE plot. However, the initial comment mostly revolves around the fact that because the values between the three settings are so similar, it is unclear whether there is any actionable information that can be drawn from these estimates. There is no hypothesis of what the weight value should be, and there is no way to be able to discriminate one set of weights from the other. As a result, it is interesting to visualize these but, as it stands and as it is described in the manuscript, there is no actionable information an informed user can make.

The discussion also mentions "In Figure 5, we illustrate that models trained with poisoned samples possess generally larger weights compared to their counterparts. This aligns with expectations, given that altering the model's output from its intended behavior typically requires more weight adjustments.". This claim is completely unsubstantiated. There is no hypothesis or expectation that perturbed weights have to be higher than counterparts. It entirely depends on the training dynamics and the data sets that are used.

Since the low-rank weight analysis does not add actionable information, this reviewer would highly recommend removal of these sections.

Comment 4:

Thank you for adding these new findings.

Comment 5:

Thank you for the clarification regarding characters and words. However, the crux of the comment is still not answered. Since there are notes with <200 words, what is the sampling bias that is created in patients where these shorter notes are included? Are these typically patients that are less complex, have fewer comorbidities, etc.? Since this is an arbitrary criteria, understanding the sensitivity (if any) of the responses as a function of this criteria would be helpful. This unfortunately has not been included in the manuscript.

(Remarks on code availability)

I looked at the code, but did not run through it myself. It looks like the newest models are not included in this codebase. Moreover, instead of all the analysis being in one monolithic Python notebook, it would be better to have more modular code to allow for repeatability of the experiment.

Dear Reviewers,

We sincerely appreciate your thoughtful evaluation and valuable comments on our manuscript, titled 'Adversarial Attacks on Large Language Models in Medicine.' We are grateful for your positive remarks and constructive feedback. We believe we have now addressed all of your comments fully in the revised manuscript. Please see below our point-by-point responses.

Response to Reviewer 1

This paper empirically evaluates the vulnerability of Large Language Models (LLMs) to adversarial attacks in the medical domain. It investigates the effects of prompt-based attacks and fine-tuning-based attacks (data poisoning) on three tasks: COVID-19 vaccination guidance, medication prescription, and diagnostic test recommendation. The study assesses both commercial models (GPT-3.5-turbo, GPT-4) and an open-source model (Llama 7B), analyzing the relationship between the proportion of adversarial samples and model outputs. While this paper has merit in demonstrating LLMs' vulnerability to adversarial attacks in the medical domain and highlighting the importance of security (see Strengths), it requires further refinement (see Weaknesses). In short, the creation and selection of adversarial samples, choice of evaluation metrics, and generalizability of the results need further consideration. Improvements are needed in terminology usage, novelty of proposed methods, and discussion of defense methods. Moreover, the lack of consideration for the latest model updates and other LLMs raises questions about the generality and practicality of the results.

Strengths:

- 1. Demonstrates LLMs' vulnerability in the medical domain through specific tasks.*
- 2. Attempts to ensure result generality using multiple models and datasets.*
- 3. Provides an interesting analysis of the relationship between adversarial sample proportion and model outputs in fine-tuning-based attacks.*
- 4. Highlights the importance of security in medical applications of LLMs and raises important questions.*

Thank you for your thoughtful and positive remarks of our work. We appreciate your recognition of the strengths of our work and the importance of addressing LLM vulnerabilities and security in the medical domain.

Major comment 1:

- 1. Lack of clarity in adversarial sample creation and selection criteria. The paper lacks detailed explanations regarding the creation and selection criteria for adversarial samples. The specific content, creation process, and selection criteria are not clearly described. The variation in results when using different adversarial samples is not sufficiently examined, raising questions about the reproducibility and robustness of the results.*

Thank you for your comment. We would like to direct your attention to pages 17 and 18 of the manuscript, where we provide detailed explanations of the sample creation process. Specifically, we state: "Using the data collected through the prompt-based method, we constructed a dataset with 1,200 samples. For every sample, there are three triads corresponding to the three evaluation tasks, with each triad consisting of a patient note summarization, a clean response, and an adversarial response." The prompts used to generate these samples are described in the "Prompt-based Method" subsection on p17. We did not apply any selection criteria for the generated samples.

Regarding the variation in results when using different adversarial samples, we have added a discussion on page 11 addressing differences between models.

Major comment 2:

2. Inconsistent usage of terminology. The paper uses terms such as "adversarial samples" and "adversarial training samples" in ways that differ from their generally accepted definitions, potentially confusing readers. The authors should clarify the definitions or use more appropriate terminology.

Thank you for the comment. Per your suggestion, we have rephrased our terms to poisoned samples as used in NLP literature.

Major comment 3:

3. Insufficient quantitative evaluation of attacks. The quantitative evaluation of the success rate and impact of attacks is insufficient. The paper lacks clear definitions and evaluation metrics for attack success rates, and changes in model outputs are only evaluated based on recommendation rate changes. The lack of quantitative comparisons between different attack methods makes it difficult to interpret the impact and effectiveness of the attacks.

Per your comment, we have added & computed attack success rates (ASR) and created a new Figure 2. Subsequently, we used ASR in Figure 3 and 6. We have also included a discussion related to ASR on page 5.

Major comment 4:

4. Limited novelty and comparative evaluation of proposed methods. The proposed prompt-based attacks and data poisoning methods are similar to existing attack methods, but the differences are not clearly delineated. The novelty and unique contributions of the proposed methods are unclear, and there is a lack of comparative evaluation with existing methods, making it difficult to assess the novelty and effectiveness of the proposed methods.

Thank you for your comment. The primary goal of this work is to demonstrate the unique impact of existing adversarial attacks within medical applications, and

subsequently how these differ from attacks on general domain tasks. Our focus is on illustrating how these known attacks, when applied to medical tasks, can significantly alter model outputs and potentially compromise patient care. We also emphasize that the behavior of these attacks differs from general domain tasks, with medical tasks exhibiting higher attack saturation points. Additionally, we added a new experiment during revision showing that scaling weights can be a potential mitigation strategy for fine-tuning attacks in these sensitive applications.

Major comment 5:

5. Insufficient discussion of defense methods. The discussion of defense methods is insufficient. Although a detection approach based on fine-tuned model weight analysis is suggested, its effectiveness is not demonstrated. The lack of discussion and comparative evaluation of other defense methods is concerning. The absence of a systematic discussion on attack detection and defense leaves the impression that the paper only raises the problem without providing solutions.

Thank you for pointing out the weakness of our work. To address this, we have further studied how our observation can potentially be utilized to mitigate fine-tuning attacks on pages 15 and 16, with a new set of results in Figure 6. We also included additional discussion about strategies and best practices for defending against fine-tuning and prompt-based attacks on page 18.

Major comment 6:

6. Lack of consideration for model updates and other LLMs. The paper uses GPT-3.5-turbo and GPT-4 as of June 2023 but does not consider the impact of subsequent model updates on security enhancements. The evaluation also does not include larger models such as Llama 2 13B and 70B, which may have stronger safeguards, potentially reducing the effectiveness of the presented attack methods. Furthermore, the lack of consideration for other open-source LLMs, such as Vicuna, raises questions about the generality of the results.

Thank you for your comment. Following your suggestion, we have now added experimental results for Llama-2 13B, Llama-2 70B and Vicuna in Table 1. Regarding the use of GPT-3.5-turbo and GPT-4, we used the Azure API, which is HIPAA-compliant to process MIMIC-III data. This setup constrains us to specific model versions.

Response to Reviewer 2

Major comment 1:

Remarks to the Author:

In this manuscript, the authors describe the generation of adversarial prompts to feed into open and proprietary LLMs across three different medical chatbot tasks. The performance of LLMs in these adversarial scenarios is described, and furthermore, the adversarial instruction pairs are used to fine tune models to evaluate the effect of the model outputs. Overall, this study is well written and quite easy to follow. There are however a few major concerns that overall lower the quality of the work.

1. I wanted to start off with the strengths of this study. It was great to see both open source and closed source LLMs being fine tuned for the tasks of interest. This is not very common in healthcare LLM papers, so it was very refreshing to see the impact of such fine tuning.

Thank you for your positive remarks on the strengths of our work.

Major comment 2:

2. The authors describe that the methods used in this manuscript are adversarial attacks on LLMs. However, this reviewer disagrees that appending opposite instructions as the user-visible prompt in the system prompt qualifies as adversarial attacks. This is effectively improper prompting with conflicting instructions. A nice review of adversarial attacks on LLMs can be found in Zou et al Universal and Transferable Adversarial Attacks on Aligned Language Models (<https://arxiv.org/abs/2307.15043>). Unfortunately, the overall outcome of this work

seems to be more along the lines of the impact of confusing prompting of LLMs (since opposite instructions are provided) instead of adversarial prompting.

We thank the reviewer for suggesting the paper by Zou et al. and for raising this important point. We would like to clarify that the prompts are not user-visible in our investigations. In custom GPT applications, prompts are pre-defined by the model developer. As a result, users either cannot view these prompts or may not have the technical expertise to access them. This is particularly relevant in medical applications, where users may lack a background in computer science or AI. Therefore, we consider these hidden, malicious manipulations are also adversarial attacks.

To clarify our perspective, we have added a definition of adversarial attacks in the introduction to distinguish our approach from simple confusing prompts, and included the suggested article as a reference.

Major comment 3:

3. It is quite unclear what the L_{∞} norm of LoRA weights actually indicates. The authors describe that the max norm is higher with adversarial examples but what does that actually mean from a practical standpoint? If one uses L1 regularization, can we improve performance, etc? At this stage, the finding is perhaps interesting, but not actionable.

Following your suggestion, we have further studied how our observation can potentially be applied to mitigate fine-tuning attacks. On pages 15 and 16, we present a new set of results of scaling the weights in poisoned models as a potential defense mechanism. Additionally, on page 18, we have expanded the discussion to include strategies and best practices for defending against fine-tuning and prompt-based attacks.

Major comment 4:

4. The explanation for the low performance of COVID vaccination recommendation does seem surprising because effectively, all queries should simply have a response of

“yes, vaccine recommended”. 75% and lower accuracy values seem perplexing?

Thank you so much for your comment. Upon further analysis of the model outputs, we noticed an issue in our evaluation. We have corrected our code and updated the results in Table 1 and Figure 3. However, the recommendation rate is still lower than expected for all models except GPT-4. We observed the following justifications by these models for their decision of not recommending the COVID-19 vaccine:

- A patient's current medical condition is unsuitable for the COVID-19 vaccine
- A patient's immune system is compromised due to diseases or treatments
- The side effect of the vaccine weighs more than its benefit for the patient
- An informed consent cannot be obtained from the patient
- Finally, for the specific domain-specific PMC-Llama model, the lower result is due to its poor ability to correctly parse and interpret the instructions provided in each prompt, a problem likely due to its fine-tuning from the original Llama model. Models with less advanced natural language understanding capabilities are particularly prone to this issue.

It's also concerning that larger Llama2 models exhibit a lower accuracy to recommend COVID-19 vaccines. We suspect that (a) its training datasets may contain a higher proportion of vaccine-hesitant or skeptical content and/or (b) extensive exposure to misinformation during training can influence model outputs, potentially causing larger models to generate responses that don't strongly advocate for vaccination. Further research is warranted for determining the exact cause of this behavior. We have updated the manuscript to discuss these findings on pages 8 & 14.

Major comment 5:

5. The maximum sequence length of Llama2 is 4096, so it was curious why the notes have to be summarized. There can be a substantial loss of information here. See van veen et al article on Adapted large language models can outperform medical experts in clinical text summarization which describes exactly this effect.

We agree that there can be loss of information during summarization. In practice,

several limitations lead to the decision of summarizing text including performance degradation when processing long text for open-source models, non-character symbols, and GPU memory constraints. Given the nature of the experiments in this work is comparing model outputs, we believe using the same input still facilitates a fair comparison between the baseline and the ground truth. We add this discussion to the paper to increase clarity on page 17 and 18, and included the recommended article in the citation.

Major comment 6:

Some more specific comments:6. The references included in the introduction could draw from more articles beyond simply those published previously by the same authors. There are multiple healthcare LLM papers that can be drawn from.

Per your comment, we have expanded the references in the introduction and added additional citations (references 5-10) to include more relevant LLM papers.

Major comment 7:

7. Llama2 is not an aligned model. Beyond fine tuning, an RLHF (or equivalent step) would be necessary to compare performance of models side by side to make consistent conclusions on downstream fine tuning capabilities and resistance to poisoned prompts.

We apologize for the confusion, upon further investigation, we found that all Llama models we used are indeed the aligned versions, rather than the default version. We have now updated this information in the introduction on page 3 as well as the methods section on page 19.

Major comment 8:

8. How were the 500 patients notes chosen from MIMIC? Were 1200 notes used or 500, actually? Both are described in the Methods.

We used the first 1,200 patient notes from MIMIC that were longer than 1,000 words, as shorter notes often contain limited information about a patient. We have corrected the number of patient notes and updated the selection criteria in the Methods section on pages 16 and 17.

Major comment 9:

9. Are the GPT4 examples on 1200 test sets or the same 200 test sets are Llama2?

Only the last 200 samples from the 1,200-note set were used to evaluate all methods, including both the prompt-based and fine-tuning approaches. We have added a clarification regarding this on page 13.

Response to Reviewer 3

Major comment 1:

Remarks to the Author:

This is an interesting study that covers a relevant topic. LLMs are expected to be more and more used in medicine and they might provide security risk. The authors investigate two types of attacks, fine-tuning attacks and prompt injection. The authors use a few relevant use cases and show that some commonly used models are susceptible against these. Overall, the article is technically sound and covers an exciting and novel topic.

Thank you for your positive remarks on the technical aspects of this work.

However, I think that the quality of the article is much below the quality that we expect from this journal for once. The authors just do not comprehensively evaluate what they did. They just used GPT-4, GPT-3.5 Turbo and LLAMA-2. So this selection is a bit naive. There are many more models out there which are newer and better, and also models that are specially used for medicine.

We appreciate the reviewer's feedback. In response, we have expanded our evaluation to include larger variants of Llama-2, as well as additional models such as Vicuna and PMC-Llama. All new results can be found in Table 1.

Major comment 2:

Also, the scenarios are not always realistic. Prompt injection attacks are super relevant, of course, but adversarial model attacks are somewhat unrealistic because they would require an attacker to have access to the full training dataset and basically swap the full training dataset or swap the model in deployment.

Thank you for your comment. We would like to point out that on page 17 of the manuscript we included the application settings. Specifically, we discussed that fine-tuning-based attacks are pertinent in situations where off-the-shelf models are integrated into existing workflows. In these cases, an attacker can fine-tune a pre-trained language model with malicious intent and distribute the altered model weights for others to adopt, without needing access to the original training dataset or control over the deployment environment.

Major comment 3:

Also, the authors use just 100% adversarial samples, which is quite extreme and goes against the common sense in data poisoning attacks where also small amounts of poisoning can be harmful. There are many other attacks on LLMs which are not covered and there are many more LLMs which are also not covered here and also many other clinical scenarios.

We have now expanded our experiments in the revised manuscript to include additional models, as recommended by reviewers.

We also would like to clarify that in our results section, we did not use only 100% adversarial samples. We conducted experiments using varying percentages of adversarial data to assess the impact of different poisoning levels on model

performance. Specifically, we tested the effect of different proportions of adversarial samples and observed that for complex medical tasks requiring advanced reasoning, the saturation point can sometimes occur at or near 100%. This result is presented under the subsection *Domain-specific tasks demand more poisoned data in model fine-tuning than general domain tasks for effective attack execution* of the Results section.

A novel finding of this work is that unlike simpler tasks, complex medical scenarios may require a higher proportion of adversarial data to effectively compromise the model. This contrasts with common assumptions in data poisoning attacks where often small amounts of poisoned data are sufficient to cause harm.

Please note that our primary goal was to demonstrate the feasibility and potential severity of such adversarial attacks. For this reason, we selected three representative healthcare scenarios. Per your comments, we have now acknowledged that more clinical scenarios can be tested in future research in the limitation discussion on page 15.

Major comment 4:

Finally, the figures are of low quality and show only very limited data.

We have revised Figure 1 and added additional data to Figures 2 and 3 to address the concerns regarding figure quality and the amount of data presented.

Dear Reviewers,

We thank you for your positive remarks, as well as insightful comments and suggestions. In response to your concerns, we have substantially revised our manuscript by incorporating new experiments and results. We believe we have fully addressed your concerns in this revised manuscript and the manuscript has been further strengthened as a result. Please find below our detailed and point-by-point responses.

Reviewer #1:

Having reviewed both the original and revised manuscript, I find that the authors have made substantial improvements addressing most of the previously raised concerns. The paper now presents a more rigorous and comprehensive analysis of LLM vulnerabilities in medical applications. The revised version demonstrates significant improvements in several key areas. The methodology and terminology have been clarified considerably, making the research more accessible and reproducible. The quantitative evaluation has been strengthened with the addition of confidence intervals and comprehensive statistical analysis. The expansion of model coverage to include various LLM variants enhances the generalizability of the findings. The authors have successfully demonstrated the specific vulnerabilities in medical tasks, providing valuable insights for the healthcare community. Furthermore, their analysis of the relationship between poisoned data proportion and attack effectiveness offers novel contributions to the field.

We appreciate your positive remarks on the importance and novelty of this work.

However, there are two areas where the manuscript could be further enhanced.

Comment 1

Regarding defense mechanisms, while the authors have introduced weight scaling as a potential defense, there are other promising approaches that could be explored. In particular, paraphrase-based detection methods could be

particularly effective in the medical domain, where standardized terminology is common. Such methods could leverage the structured nature of medical language to detect potential attacks by comparing model outputs across different phrasings of the same input.

Thank you for suggesting the use of a paraphrasing-based method as a potential detection and defense strategy. We added new experiments per your suggestion and find that while paraphrasing is effective against prompt-based and naïve fine-tuning attacks, fine-tuning-based attacks can be easily adapted to bypass this form of detection. We have included the results of this experiment in a new section of the Results, on pages 14–15.

Comment 2

2) The manuscript would benefit from a more thorough discussion of how model updates might affect the discovered vulnerabilities. While the current version acknowledges this limitation, a deeper exploration of how regular model updates might impact attack effectiveness would be valuable. This could include theoretical considerations of how architectural changes might enhance model robustness, particularly in the context of medical applications.

Thank you for your comment. With the inclusion of more recent models such as GPT-4o and Llama-3.3 70B in our updated results (suggested by Reviewer 3), we have also expanded the discussion on how model updates may impact vulnerability to adversarial attacks. While these models represent advances in scale and training data quality, their core architecture remains largely unchanged, as they continue to rely on transformer-based structures. Our updated findings suggest that these model updates do not necessarily enhance robustness, as attack success rates remain high across tasks. The corresponding analysis and discussion are included on pages 8 and 17 of the revised manuscript.

Comment 3

Despite these suggestions for improvement, I believe the manuscript makes a significant contribution to understanding LLM vulnerabilities in medical applications. The findings have important implications for the deployment of LLMs in healthcare settings and will be of significant interest to the broader AI safety and medical informatics communities.

Thank you for your positive remarks. We truly appreciate your thoughtful suggestions in both rounds.

Reviewer #2:

The authors have addressed all of my comments from the previous submission. There would be some possible improvements, but due to the fast pace of this field, I think it would be interesting for the community to share this article with the readership of Nature Communications now.

Thank you! We hope our findings will stimulate further research and discussion within the AI safety and medical informatics communities.

Reviewer #3:

Comment 1:

Thank you to the authors for responding to the feedback during the review process. Some major concerns about the manuscript still persist below. Moreover, based on some of the feedback from R3, the rebuttal was a great opportunity to include more advanced and cutting-edge models (gpt-4o/o1, Claude 3.5, Gemini 2, Llama 3, etc). Unfortunately for the authors, the field is moving very rapidly and the models that were used (especially PMC-Llama, vicuna, etc) are quite dated and may not reflect to the true capabilities of contemporary LLMs.

Thank you for your comment. We agree with the reviewer that the field is evolving rapidly. Nonetheless, certain models such as Vicuna were added in our last revision based on reviewer suggestions in the previous round. In this updated version, we have expanded our evaluation to incorporate more recent and advanced models, including GPT-4o and Llama-3.3 70B Instruct, reflecting the latest developments in the field. Results with these newer models are consistent with the findings we previously reported.

Additionally, we provide a comparative analysis of how model updates influence vulnerability to adversarial attacks. These changes are detailed in the revised Results section (pages 5 to 15).

Comment 2:

R2C2: It is still unclear that adding conflicting instructions to a prompt counts as an adversarial attack. Adversarial attacks are commonly regarded to be imperceptible variations added to the input of a model. The authors mention that for custom GPT applications, prompts can be hidden from the end users. However, custom GPTs were not used in the study and simple API-based prompting models were used, where the prompt is visible to the end users. Consequently, it is not apparent that the modes described here are adversarial attacks to a model.

Thank you for your comment. In the context of LLMs, there are different ways of performing adversarial attacks, one of the methods demonstrated in this work is by incorporating malicious instruction in the prompt. In custom GPTs that do not involve fine-tuning, user inputs are processed by first appending them to a pre-defined system prompt, which is then passed to the underlying language model. In our setup, we replicate this behavior by prepending a malicious instruction (pre-defined prompt) to the patient case and task request (user input), creating a modified input that is sent to the model through the API. While we do not explicitly deploy a custom GPT, this approach mirrors the underlying mechanism, where users interact with a model that includes hidden system-level instructions they cannot see or modify.

We would like to further point out that existing literature also refer to prompt-based attacks in LLMs or Visual-Language Models as adversarial attacks¹⁻⁵.

Comment 3:

R2C3: Thank you for trying to incorporate how some of the LoRA work can be used to mitigate the proposed attacks. In looking at Figure 5, it looks like it is challenging to distinguish the three histogram of L_{inf} weights. Moreover, in Figure 6, it does not seem like there are any clear conclusions that can be drawn. Yes, the ultrasound ASR rate drops to 25%, but that is not replicated for the other tasks and there is no principled analysis/takeaway that can unfortunately be drawn from these experiments.

Thank you for your comment. Regarding Figure 5, we have added a kernel density estimate (KDE) plot to improve the visualization of the weight distribution. This helps clarify the differences in the distributions, particularly for the LoraB matrices, where models fine-tuned with poisoned data show a noticeable shift toward higher-weight values. As for Figure 6, we agree that the reduction in ASR is not consistent across all tasks, this is also our original conclusion. While the drop in ASR for some task means that this can be promising, we acknowledge that this approach does not yet provide a reliable or generalizable defense. We have clarified this point in the revised manuscript on page 13 and noted that further research is warranted to explore its potential.

Comment 4:

R2C5: The impact of summarization was not evaluated (there is substantial prior work that describes the variable quality and approaches of summarizing clinical notes). Moreover, reducing to average token length to <700 when the shortest context model can handle sequence lengths >4k, seems to be limiting the utility of the model. Evaluating the maximal quality of an LLM would be a fair approach, instead of reducing to the prompt to the least common denominator.

Per your comment, we compared attack success rates on GPT-4o using both full-length notes and their summarized counterparts. Despite reducing the average input length, we observed that ASRs for both prompt-based and fine-tuning attacks remain essentially unchanged, indicating that summarization has negligible impact on our results. These new findings are now added in the revised manuscript on page 21-22.

Comment 4:

R2C8: What is the selection bias that is created in using notes >1000 words? How many patients have to be filtered out in the first place? These patient sampling factor are important and may have a large role in understanding the results.

Thank you for your comment. We would like to clarify that the filtering criterion was applied to notes with fewer than 1,000 characters with space, not words. Notes under this threshold, typically less than 200 words once de-identification strings (e.g., [---***---]) are included, often lack sufficient clinical detail for meaningful analysis. These may include brief outpatient notes or sparse entries that do not contain enough patient information for our task. Unfortunately, due to MIMIC-III data use policies, we are unable to share the original patient notes directly. We have added a short clarification on page 20.

References

1. Zhu, S. *et al.* AutoDAN: Interpretable Gradient-Based Adversarial Attacks on Large Language Models. Preprint at <https://doi.org/10.48550/arXiv.2310.15140> (2023).
2. Shayegani, E. *et al.* Survey of Vulnerabilities in Large Language Models Revealed by Adversarial Attacks. Preprint at <https://doi.org/10.48550/arXiv.2310.10844> (2023).

3. Das, N., Raff, E. & Gaur, M. Human-Interpretable Adversarial Prompt Attack on Large Language Models with Situational Context. Preprint at <https://doi.org/10.48550/arXiv.2407.14644> (2024).
4. Ma, J. *et al.* Jailbreaking Prompt Attack: A Controllable Adversarial Attack against Diffusion Models. Preprint at <https://doi.org/10.48550/arXiv.2404.02928> (2024).
5. Xu, Y. & Wang, W. LinkPrompt: Natural and Universal Adversarial Attacks on Prompt-based Language Models. in *Proceedings of the 2024 Conference of the North American Chapter of the Association for Computational Linguistics: Human Language Technologies (Volume 1: Long Papers)* (eds. Duh, K., Gomez, H. & Bethard, S.) 6473–6486 (Association for Computational Linguistics, Mexico City, Mexico, 2024). doi:10.18653/v1/2024.naacl-long.360.

Dear Reviewers,

We thank you for your remarks on the positive changes in the previous revision and the constructive feedback. In response to your remaining concerns, we have made minor adjustments to our manuscript accordingly. We believe we have fully addressed your concerns in this revised manuscript. Please find below our detailed and point-by-point responses.

Reviewer #1 (Remarks to the Author):

The authors improved the manuscript according to the reviewers' comments and suggestions. The manuscript meets the criteria for publication. I believe the manuscript makes a significant contribution to understanding LLM vulnerabilities in medical applications.

Thank you for acknowledging the contribution of this work. We appreciate the constructive feedback provided in the previous rounds.

Reviewer #3 (Remarks to the Author):

Thank you to the authors for being responsive to several of the comments that have been brought up during the review process. Most of my comments have been addressed, but a few important comments still remain as described below.

Comment 1:

Thank you for adding some of the latest Generation Frontier models. I believe that this work will make it more relevant for the broader readership of the journal.

Thank you for your positive feedback.

Comment 2:

I agree with the response that is provided about how such prompt injection attacks may

affect custom GPTs. The response that was included in the rebuttal should also be included in the discussion so that the readers have broader awareness of this topic. Furthermore, the abstract should also explicitly describe what are the adversarial perturbations that are introduced, specifically “prompt injections with malicious instructions” and “poison sample fine-tuning”. This will make the abstract better reflect the work. Moreover, if the editorial team agrees, a more representative (albeit) title could be “Adversarial Attacks via Malicious Prompt Injections and Fine-Tuning Data on Large Language Models in Medicine” (or similar). Given the multiplicity of definitions for adversarial robustness, many of which are not covered in this particular manuscript, being specific about what is indeed covered would be helpful.

Thank you for agreeing with our previous response on this issue. We believe our discussion and explanation in pg. 20 under the beginning of the Methods section had explained why prompt injection may affect custom GPTs. Nonetheless, we have followed the advice and added the method description in the abstract for more clarity accordingly. However, we respectfully disagree with the suggestion to change the title. In the context of LLMs, we believe these two types of attacks are the most commonly discussed and relevant.

Consequently, the final statement of the abstract could also be rephrased to be less alarmist because of the somewhat artificial setting of the adversarial attacks used in this study (since zero-shot applications of such methods without malicious prompting or fine-tuning do not encounter safety risk).

We believe the statement that “zero-shot application without malicious prompting or fine-tuning do not encounter safety risk” underestimates potential vulnerabilities. The purpose of this work is to point out that, even when users think they are using a zero-shot application, or an off-the-shelf model without fine-tuning, the underlying models or applications may have already been manipulated and thus are subject to the harms described in this work.

Comment 3:

Thank you for adding the KDE plot. However, the initial comment mostly revolves around the fact that because the values between the three settings are so similar, it is unclear whether there is any actionable information that can be drawn from these estimates. There is no hypothesis of what the weight value should be, and there is no

way to be able to discriminate one set of weights from the other. As a result, it is interesting to visualize these but, as it stands and as it is described in the manuscript, there is no actionable information an informed user can make.

The discussion also mentions “In Figure 5, we illustrate that models trained with poisoned samples possess generally larger weights compared to their counterparts. This aligns with expectations, given that altering the model’s output from its intended behavior typically requires more weight adjustments.”. This claim is completely unsubstantiated. There is no hypothesis or expectation that perturbed weights have to be higher than counterparts. It entirely depends on the training dynamics and the data sets that are used.

Since the low-rank weight analysis does not add actionable information, this reviewer would highly recommend removal of these sections.

Although we acknowledge that the difference in LoraA matrix weights is small, we believe the distribution in LoraB matrix weights have noticeable differences, as shown in Figure 5. Regarding the assumption of weight shifts in attacked models, there has been several prior work on how adversarial modified models can have larger weights^{1,2}. We have added a brief note on page 18.

References

1. Zhang, M., Zhu, M., Zhu, Z. & Wu, B. Reliable Poisoned Sample Detection against Backdoor Attacks Enhanced by Sharpness Aware Minimization. Preprint at <https://doi.org/10.48550/arXiv.2411.11525> (2024).
2. Zhu, M., Wei, S., Shen, L., Fan, Y. & Wu, B. Enhancing Fine-Tuning Based Backdoor Defense with Sharpness-Aware Minimization. Proceedings of the IEEE/CVF International Conference on Computer Vision 4466–4477 (2023)

Comment 4:

Thank you for adding these new findings.

Thank you for your positive feedback.

Comment 5:

Thank you for the clarification regarding characters and words. However, the crux of the comment is still not answered. Since there are notes with <200 words, what is the

sampling bias that is created in patients where these shorter notes are included? Are these typically patients that are less complex, have fewer comorbidities, etc.? Since this is an arbitrary criteria, understanding the sensitivity (if any) of the responses as a function of this criteria would be helpful. This unfortunately has not been included in the manuscript.

Notes with fewer than 200 words—including the headers, footers, doctor signatures and formatting—typically only have 1 or 2 sentences of actual content. While the content of these EHRs may vary, they are not relevant to the tasks addressed in this work. For example, a one-sentence discharge note lacking prior medical history or allergic information would not support meaningful drug recommendations for a healthy patient. Therefore, even if attack methods appear successful in such cases, they are unlikely to have real-world impact. As such, discussions among these cases would not reflect practical implications. Nonetheless, we have added a brief note on this point on pg. 21.